# Risk for Surgery in Patients with Polyposis Syndrome after Therapy by Device-Assisted Enteroscopy (DAE): Long-Term Follow Up

**DOI:** 10.3390/jcm11040899

**Published:** 2022-02-09

**Authors:** Clelia Marmo, Annalisa Tortora, Guido Costamagna, Rebecca Nicolò, Maria Elena Riccioni

**Affiliations:** 1UOC Endoscopia Digestiva Chirurgica, Fondazione Policlinico Universitario A Gemelli IRCCS, Università Cattolica del Sacro Cuore, 00168 Rome, Italy; guido.costamagna@unicatt.it (G.C.); mariaelena.riccioni@unicatt.it (M.E.R.); 2UOC Gastroenterologia e Medicina Interna, Fondazione Policlinico Universitario A Gemelli IRCCS, Università Cattolica del Sacro Cuore, 00168 Rome, Italy; annalisa.tortora@policlinicogemelli.it; 3UOC Medicina d’Urgenza e Pronto Soccorso, Fondazione Policlinico Universitario A Gemelli IRCCS, 00168 Rome, Italy; rebecca.nicolo@policlinicogemelli.it

**Keywords:** enteroscopy, polyposis, small bowel disorders

## Abstract

Background and aim of the study: Polyposis syndromes such as Peutz–Jeghers (PJ) and familial adenomatous polyposis (FAP) are associated with the growth of small bowel polyps; the risk is approximately 60–90% for PJ and 40–70% for FAP. The primary aim of this study was to evaluate the efficacy of device-assisted enteroscopy (DAE) in the detection and treatment of small bowel polyps to reduce the risk of surgery. The secondary objective was to study complications and mortality. Methods: We conducted a retrospective cohort study by analyzing a structured database. Between September 2006 and October 2019, we observed and followed 42 consecutive patients with polyposis syndromes; they underwent device-assisted enteroscopy and three were excluded from elective surgery after the exam. The endoscopic exams were performed for diagnostic and therapeutic purposes. Results: Thirty-nine patients were evaluated with a mean follow up of 6.7 years (±SD 2.7), 79.5% were female with a mean age of 43.8 years (±SD 15.02), and 68 enteroscopies were performed with the removal of 64 polypoid lesions. One bleeding episode occurred after operative enteroscopy, and the need for subsequent surgery occurred in six patients with PJ and in five patients with FAP. The surgical indications in PJ patients were the presence of large polyps (three patients) and three cases of intussusception, one of which was a patient with a polyp in the proximal ileum, not reachable with the scope. One patient with PJ died from pancreatic cancer during follow up. The surgical indications in patients with FAP were the presence of four large polyps with high-grade dysplasia and one ampullary neoplasia recurrence. Conclusions: In PJ patients, the endoscopic treatment of small bowel polyps was safe. During the follow-up period, the patients with successful endoscopic treatment did not need surgery. In FAP patients treated with DAE, none developed cancer.

## 1. Background

Polyposis syndromes such as Peutz–Jeghers (PJ) and familial adenomatous polyposis (FAP) are inherited disorders characterized by the growth of polypoid lesions in the gastrointestinal tract [1].

Peutz–Jeghers syndrome is an autosomal dominant disease associated with polyposis in the gastrointestinal tract and mucocutaneous pigmentation, especially on the lips and inside the mouth [1]. The germline mutation is on the chromosome 9p13.3, encoding a serine-threonine kinase, STK11. This gene acts as a tumor suppressor, and its loss of heterozygosis is correlated with an increased risk of developing cancer in the gastrointestinal tract and breast cancer [2].

The polyps of the gastrointestinal tract have a characteristic histology, defined as hamartomatous. They show a typical elongated epithelial structure associated with a dilated cystic gland that extends in the submucosa or in the muscolaris propria, and smooth muscle is found in the polyp projections. Another typical feature of this type of polyp is a long stalk. Its origin is controversial, but the stalk is probably generated by the mechanical propulsion of peristalsis associated with an abnormal growth of the stromal component. Polyps are mostly found in the small bowel (60–90%) and in the colon (50–64%) with clinical presentation in childhood (33%) and by 20 years of age (50%) [1]; despite follow up often being performed in an affected family, the clinical onset frequently consists of a bowel obstruction.

Recently, the European Society of Gastroenterology (ESGE) developed clinical guidelines for the diagnosis and management of this condition [3]. After the diagnosis was made, the ESGE recommends performing a baseline esophagogastroduodenoscopy and a colonoscopy for risk stratification at age 8. If the baseline exams are negative, the surveillance starts at age 18. If there are positive findings, the aforementioned exams must be repeated every 1–3 years based on the phenotype of the disease, starting at age 8. The same recommendation is provided for small bowel surveillance, using video capsule enteroscopy or MRI as baseline exams [3].

The guidelines suggest removing polyps larger than 15–20 mm in the small bowel and in the colon to prevent the future need for surgery. For small bowel polyps, the indication is a resection during device-assisted enteroscopy (DAE) and intraoperative enteroscopy for larger polyps and for patients with multiple polyps [3]. 

Familial adenomatous polyposis (FAP) is an autosomal dominant disorder associated with a germline mutation in the APC gene situated on chromosome 5q21 [4]. This gene acts as a tumor suppressor or ‘gatekeeper’ and its alteration is involved in the development and progression of colorectal cancer. Several mutations are acknowledged as responsible of this genetic disease and there are heterogeneous genotypes and phenotypes for the same mutation [5]. 

Current guidelines [3] suggest starting colonoscopy surveillance in asymptomatic individuals with familial adenomatous polyposis at the age of 12–14 years; in patients with an intact colon, a follow-up colonoscopy every 1–2 years is recommended. This strategy allows to reduce the incidence of colorectal cancer (CRC) and related mortality, and optimizes the timing of surgery. All polyps detected that are 5 mm or more should be endoscopically removed, though there is no indication of management with endoscopic polypectomy alone. Guidelines suggest prophylactic surgery as the standard of care: colectomy or proctocolectomy with ileo-pouch anal anastomosis [3]. 

The duodenum is often affected by malignant lesions in patients with FAP; duodenal adenomas and cancer are common, as well as adenomas of the papilla and perivaterian region [6,7]. The Spigelman score [8] is useful in the stratification of risk in the duodenum and predicts the appropriate interval for duodenal surveillance. 

The guidelines suggest starting endoscopic duodenal surveillance at the age of 25 and endoscopically removing all lesions that are less than 10 mm in the duodenum. For larger adenomas, there are no recommendations for endoscopic removal due to the increased risk of invasive growth and relapse [3]. 

In the jejunum, lesions are less frequent; therefore, there is no strong recommendation to study the small bowel in these patients. However, an increased risk of developing a jejunal adenoma or cancer is described in individuals with FAP [9]. The American Society of Gastrointestinal Endoscopy (ASGE) guidelines [10] recommended evaluating the small bowel, with capsule endoscopy or magnetic resonance imaging, in patients with a higher risk of small bowel cancer (Spiegelman score IV) and the endoscopic resection of small bowel polyps with deep or device-assisted enteroscopy when needed.

## 2. Aim

The primary aim of this study was to evaluate the efficacy of device-assisted enteroscopy in the detection and treatment of small bowel polyps to reduce the risk of recurrent surgery in patients with Peutz–Jeghers syndrome and familial adenomatous polyposis. 

The secondary aim was to evaluate the adverse events related to the procedure and mortality for all causes.

## 3. Methods

### 3.1. Patients Selection

This study was conducted in a tertiary referral center, analyzing a structured database organized during the period from September 2006 to October 2019. Participants were adult patients aged over 18 and affected by PJ syndrome or FAP with documented or suspected small bowel lesions. Exclusion criteria included age under 18 years, pregnant or breastfeeding women, patients with known or suspected intestinal obstruction or perforation, contraindications to enteroscopy and lack of consent to the endoscopic procedure. The structured database collected patients’ demographic data, indication for procedure, patients’ history, type of treatment and the results of the endoscopic treatments. Patients were initially subjected to radiological analysis (magnetic resonance imaging or computer tomography) or underwent video capsule endoscopy to study the small bowel. 

### 3.2. Patient Preparation and Endoscopic Intervention

All patients underwent device-assisted enteroscopy (DAE). Endoscopic exams were performed with a single balloon enteroscopic system, SIF-Q180 (Olympus Optical Co., Tokyo, Japan), or a spiral motorized enteroscope (Spirus Medical LLC, West Bridgewater, MA, USA), which in some cases were under fluoroscopic control for therapeutic indications. Patients were admitted with ordinary hospitalization or a day hospital regimen, based on previous small bowel examination of PJ patients, while for FAP patients, we performed both oral and anal examinations due to the risk of cancer. For oral exams, abstention from solid food for 12 h and from liquids for 4 h was prescribed before proceeding. For anal exams, 4 L of polyethylene glycol (PEG) were administered the afternoon before in case of a procedure programmed for the morning, and in split dose (2 L in the evening the day before and 2 L in the morning of the same day) for procedures programmed in the afternoon. A low-fiber diet was prescribed in both cases for 48 h before the endoscopic exam. The endoscopic exams were performed with diagnostic and therapeutic purposes and all were conducted in deep sedation with anesthesiologic assistance and using CO_2_ inflation. During the polypectomy, we obtained complete resection with a snare or the apposition of an endoloop or endoclips on the stalk to induce the necrosis of the polyp. The snares we used were Captivator Single-Use Snares (10–15–25–37 mm) (Boston Scientific Corporation, Middlesex County, MA, USA), Olympus snares (10–15–25 mm) (Olympus Medial System Corp.) or an oval monofilament mucosectomy snare (Meditalia). The electrosurgery was with ERBE VIO 2 (Erbe Elektromedizin GmbH, Tuebingen, Germany) with an effect of cut 2 and forced coagulation 30 Watt. The endoclips used were repositionable clips: hemostatic clips (Micro-Tech, Co. Ltd., Nanjing, China) and resolution clips (Boston Scientific Corporation, USA). Whenever possible, the large polyps were recovered with a basket (Roth-net, STERIS, US Endoscopy) and the smaller ones were sucked in a polyp trap for histological analysis. The diagnosis and treatment of the polyps was performed during the withdrawal of the scope and following the intravenous administration of antispasmodic drugs (N-scopolamine or glucagon). The choice to position a loop-ligating device, such as an endoloop (Olympus Medial System Corp.), or clips on the stalk of the polyps was adopted for patients with Peutz–Jeghers syndrome due to the benign nature of the polyps and its very low risk of cancerization. We used a standardized classification in order to distinguish polyps by size:-small (<10 mm);-medium (>10 mm and <20 mm);-large (>25 mm).

After the procedures, patients were monitored in a hospital ward for 24 h for the more invasive therapeutic exams and for 4 h in a day hospital regimen for the simpler therapeutic procedures.

The endoscopic treatment was considered a success if the polyps were removed or if the enteroscopy was propaedeutic to surgery (tattoo or indication for elective surgery). The endoscopic treatment was considered a failure when patients underwent emergency surgery.

### 3.3. Statistics

A descriptive analysis of the data was performed, with a calculation of the mean, median, proportions, standard deviation and 95% confidence limits depending on whether they were continuous, ordinal or nominal values. The analysis of inference was performed with univariate tests (*t*-tests, Mann–Whitney tests, χ^2^ tests) depending on the type of variables analyzed. Odds ratios were calculated with 95% confidence limits. All calculations were made with the STATA package 15.1 (StataCorp LP, College Station, TX, USA).

## 4. Results

Among the 42 patients enrolled, 3 were excluded because of the indication for elective surgery after enteroscopy, i.e., the patients of the first two enteroscopies performed in our institute were not treated endoscopically due to the little experience we had at the time on small bowel polypectomy technique. 

A total of thirty-nine patients were evaluated with a mean follow up of 6.7 years (SD 2.7); 79.5% were female, with a mean age of 43.8 years (±SD 15.02). Moreover, 86.6% of PJ patients and 100% of FAP patients had previous surgery. The indications in PJ patients were intussusception or intestinal obstruction, whilst in FAP patients, the indication was colorectal cancer. The characteristics of the patients are described in Table 1. 

Overall, sixty-eight enteroscopies were performed, including via the oral and anal routes, as shown in Table 2. All procedures were performed on the base of previous examinations that showed or raised the suspicion of a small bowel lesion. Twenty-four patients underwent therapeutic procedures to treat 65 polypoid lesions, mostly classified as small polyps (70.7%), followed by large and medium polyps (17% and 12.3%, respectively) Table 3.

Surgery was necessary in eleven patients (28.9%)—namely in six patients with Peutz–Jeghers syndrome (PJ) and in five patients with familial adenomatous polyposis (FAP). The surgical indications in PJ patients were the presence of large polyps over 30 mm of diameter with a long stalk and a big head which we classified as not safely removable, three intussusceptions, two missed by small bowel capsule endoscopy and device assisted enteroscopy, and one patient with a known polyp in the proximal ileum not reachable by enteroscope. The surgical indications in patients with FAP were (see Table 4 for details): four large polyps with high-grade dysplasia in the duodenum and proximal ileum and one recurrence of ampullary neoplasia. No cases of death were observed after the endoscopic procedures or after surgery. We reported, during the follow up, one case of death due to pancreatic cancer in a patient with PJ at the age of 45 years.

The procedures were safe; one major complication was observed, i.e., a bleeding episode occurred after the removal of two large polyps of the jejunum in a PJ patient needing one packed red blood cell transfusion with the complete remission of the patient. 

## 5. Conclusions

In this cohort of patients with polyposis syndromes, our intention was to evaluate the role of the device-assisted enteroscopy to reduce the risk of recurrent surgery. In the Peutz–Jeghers subgroup, three patients required emergency surgery following obstruction due to intussusception, all three after therapeutic enteroscopy that did not reach the target lesion. Three more patients were operated after diagnostic enteroscopy that reached the lesion and indicated surgery as the best option for the removal of polyps in consideration of their large size. In the familial adenomatous polyposis subgroup, the indication for surgery was always made following a previous enteroscopy that provided diagnostic information about the site, size and type of polyp to remove. Currently, guidelines [3,10] suggested searching for and removing polyps in the small bowel, even if small in size, in order to reduce the risk for intussusception and bleeding in patients with Peutz–Jeghers syndrome and for cancer prevention in patients with familial adenomatous polyposis. Little is known about which is the most effective method to explore the small bowel between capsule endoscopy or device-assisted enteroscopy. A recent metanalysis compared 15 studies, overall including 821 patients, on small bowel examination with capsule endoscopy, device-assisted enteroscopy or both, for polyposis syndromes [11]. This study has shown a similar effectiveness of the two procedures in detecting polyps or cancer in the small bowel [11]. However, the capsule endoscopy has a higher number of complete examinations when compared with device-assisted enteroscopy [11]. It must be taken into consideration that device-assisted enteroscopy has therapeutic possibilities; therefore, it is reasonable to hypothesize the combined use of the two methods to detect and treat small bowel lesions. Other authors reported the effectiveness of device-assisted enteroscopy in removing polyps in the small bowel, without a consistent risk for the patients. The polyps that we removed were more often localized in the proximal part of the jejunum or in the distal ileum, with varying sizes up to 6 cm in one case. Sometimes more than one session was required to eradicate the detected polyps [11,12,13]. The procedure also seems to be safe in the pediatric population [14].

One patient with PJ syndrome in our cohort developed pancreatic cancer during the follow-up period. The risk of extra gastrointestinal cancer in PJ patients was previously described [15,16,17]; in these analyzed studies, the risk of pancreatic cancer was approximately 5% at the age of 50 years. 

In terms of our study’s limits, this was a retrospective study and almost all patients had previously undergone surgery before coming to our attention; this population has probably already been selected as at risk of surgery, selection bias, and lacks a control group. Failure of endoscopic removal was related to the size of the polyps or to a deep site not easily accessible due to previous surgery, as reported by other authors [12,18,19]. 

In our experience, with a follow-up period of 6.7 years, 71% of patients with polyposis syndromes avoided abdominal surgery. In PJ patients, the indication for elective surgery was given following endoscopic exams and in the case of treatment failure. In regard to cancer prevention, endoscopic exams are effective: no patient developed a gastrointestinal cancer in our study cohort during the follow up.

## Figures and Tables

**Table 1 jcm-11-00899-t001:** Patients’ demographic characteristics.

Factors	Mean Years SD
Age	43.8 SD ± 15.0
Overall Follow Up	6.71 SD ± 2.8
**Factors**	**Nr. (%) [C.I. 95%]**
Female	31 (79.5%) [64.5–89.2]
Peutz–Jeghers	15 (38.5%) [24.9–54.1]
FAP	24 (61.5%) [45.9–5.1]
**Previous Surgery in PJ**	**13 (86.6%) [C.I. 62.1–96.2]**
Small Bowel Resection	11
Colic Resection	2

**Table 2 jcm-11-00899-t002:** Route of insertion, indication and enteroscopic procedure.

Type of Enteroscopy	Nr. [C.I. 95%]
Single Balloon	64 (94.1%) [C.I. 85.8–97.7]
Spiral	4 (5.9%) [C.I. 2.3–14.2]
**Indication**	
Therapeutic	24 (35.3%) [C.I. 25–47.16]
Diagnostic	44 (64.7%) [C.I. 52.8–75]
**Route of Insertion**	
Oral	53 (77.9%) [C.I. 66.7–86.1]
Anal	15 (22.1%) [C.I. 13.8–33.3]

**Table 3 jcm-11-00899-t003:** Morphological characteristics of the removed polyps.

Polyps’ Size	Nr. 65 (%) [C.I. 95%]
Small nr. (%)	46 (70.7%) [C.I. 58.8–80.4]
Medium nr. (%)	8 (12.3%) [C.I. 6.4–22.5]
Large nr. (%)	11 (17%) [C.I. 9.72–27.8]

**Table 4 jcm-11-00899-t004:** Indications for surgery after enteroscopy.

Need for Surgery	Nr. (%) [C.I. 95%]
Emergency	3 (7.9%) [C.I. 2.7–20.8]
Elective	8 (21%) [C.I. 11–36.3]
Overall	11 (28.9%) [C.I. 17–44.8]
**Indication for Surgery**	(%) [C.I. 95%]
Peutz–Jeghers nr. (%)	6/15 (40%) [C.I. 19.8–64.2]
Familial Adenomatous Polyposis nr. (%)	5/24 (20.8%) [C.I. 9.2–40.5]

## Data Availability

The data presented in this study are available on request from the corresponding author.

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
