# Peer review of "Risk for Surgery in Patients with Polyposis Syndrome after Therapy by Device-Assisted Enteroscopy (DAE): Long-Term Follow Up"

_jcm, 2022, doi:10.3390/jcm11040899_

Round 1
Reviewer 1 Report
In this retrospective study the Authors evaluated the efficacy of Device Assisted Enteroscopy in detection and treatment of small bowel polyps to reduce the risk of surgery and, secondly, the mortality and the complication rates in patients with Polyposis syndromes like Peutz-Jeghers (PJ) and Familial Adenomatous Polyposis (FAP).
Out of 42 patients recruited between September 2006 and October 2019, thirty-nine patients (15 with PJ and 24 with FAP) were treated by endoscopical approach whilst three were excluded from elective surgery. Overall, sixty-eight enteroscopies were performed and sixty-five polypoid lesions were removed.
During a follow-up of 6.7 years a successful endoscopic treatment, without need surgery, was reported in twenty-eight patients whilst the need for subsequent surgery occurred in sis patients with PJ and in five patients with FAP.
One bleeding episode occurred after operative enteroscopy whilst one patient with PJ died from pancreatic cancer during the follow up.
On bases of these results endoscopic treatment of small bowel polyps believed be safe and effective in reducing the risk of surgery. Its retrospective design and the lack of a control group.
These limits would be underlined in discussion.
Author Response
"Its retrospective design and the lack of a control group. These limits would be underlined in discussion."
I underline the limits and the text is modified:
"...Limits, this is a retrospective study and almost all patients had a previous surgery before coming to our attention; probably it was a population already selected as at risk of surgery, selection bias, and it lacks of control group. The failure of endoscopic removal is related to the size of polyps or to a deep site not easily accessible due to previous surgery, as reported by other authors (12) (18) (19)..."

Reviewer 2 Report
The manuscript has been improved, however the tables need to be further improved since the CI listed pose no value to the reader, and the data should be presented in a more concise type of way. The Results should only serve to list the results not justify why certain patient should not be considered treatment failure (rather, list the criteria for treatment failure you have used). Also, the introductory sentence to the Discussion is not in line with the aim nor the study results.
Author Response
- "the tables need to be further improved since the CI listed pose no value to the reader, and the data should be presented in a more concise type of way". I have modified the tables in a more clear form.
- "The Results should only serve to list the results not justify why certain patient should not be considered treatment failure (rather, list the criteria for treatment failure you have used)". I have removed this information from the results and reported it in the methods section.
- "The introductory sentence to the Discussion is not in line with the aim nor the study results". I have changed the sentence as reported below: "...In this cohort of patients with polyposis syndromes, our intention was to evalutate the role of the device assisted enteroscopy to reduce the risk of recurrent surgery...".

Round 2
Reviewer 2 Report
Thank you very much for the changes made to the manuscript.
This manuscript is a resubmission of an earlier submission. The following is a list of the peer review reports and author responses from that submission.
Round 1
Reviewer 1 Report
The article should be revised in order to be accepted, the specific comments are listed below.
Abstract: In the abstract all abbreviations should be first explained, and used later, for instance DAE. Once the abbreviation has been established in the main body of the manuscript, it does not need to be repeatedly explained.
Introduction: Introduction is too long, it should not have subsection and highlights (instead of full sentences) should not be used. Even more important, references must be inserted in cases when the authors are stating societal guidelines or in cases when they are presenting scoring system derived by other authors. Aims section should be shortened to a single sentence, and objectives going into more details should be listed as a part of Methods.
Methods: The period of thirteen years is an unspecific study period, especially when the authors themselves have elaborated in the Results that the first patients originally included in the study, later on needed to be excluded due to the fact they were not treated endoscopically. Moreover, more details should be provided on the endoscopy equipment and statistical program used. Once again, full sentences should be used even when the authors are enumerating.
Results: Tables should be revised to be more visually appealing and easier to follow. Nothing regarding the risk of surgery is at this time described in the Results deeming the title of the Manuscript obsolete.
Conclusions: References are missing. The table should be omitted and explained textually. Last sentence does not seem to be in direct link to the manuscript topic and should be omitted.
Reviewer 2 Report
Thank you for the privilege of reviewing your work.
It is an original work, well written and well presented.
Nevertheless a few comments:
- It is not explained when and why the patients were reevaluated (39 patients- 68 enteroscopies)
- It is mentioned that 37/39 patients had previous small or large bowel resection- what was the indication for these resections?
- The conclusion is not safe- you are fairly rising the subject of small bowel surveillance in FAP and Peutz-Jeghers patients- but your results cannot safely support your conclusions. (pporly defined neoplasia history- not long enough follow-up)
- Limitations of the study should be reported
- Is this an IRB approved database set? Thank you,